# MIND-MAP AGENT: ENHANCING COOPERATIVE TASK PLANNING THROUGH COMMUNICATION ALIGNMENT WITH LARGE LANGUAGE MODELS

## ABSTRACT

Embodied agents that collaborate with humans through natural language have become an active area of research, offering flexibility in cooperative planning and execution. Debate-based approaches often depend on repeated consensus procedures, which can increase dialogue frequency and risk over-communication. At the same time, LLMs are prone to hallucination during dialogue processing, sometimes causing confabulation and reducing consistency in long-term strategies. We introduce the Mind-Map Agent, an approach that guides reasoning with explicit cooperative strategies while maintaining structured long-term memory to disentangle dialogue, task state, and planning context. The generated Mind-Maps support coherent long-horizon planning, reduce redundant dialogue, and enhance interpretability in multi-agent interaction. Evaluations on Communicative Watch-and-Help and ThreeDWorld Multi-Agent Transport indicate that the Mind-Map Agent achieves more stable efficiency compared to classical planners and LLM agents across different model scales and environments. Our results suggest that Mind-Map reasoning enables cooperative agents to accomplish tasks with fewer conversations while sustaining effective collaboration.

## 1 INTRODUCTION

Recent advances in Large Language Models (LLMs) have enabled new paradigms in multi-agent embodied tasks, particularly in natural language–based reasoning, planning, and decision-making Brohan et al. (2023); Jain et al. (2020). Traditional multi-agent systems, while reliable under predefined communication protocols Stone & Veloso (2000), often struggle in decentralized collaboration where agents must infer teammates' intentions Chang et al. (2025); Zhang et al. (2024) and dynamically coordinate without centralized control. The challenge becomes even more pronounced in human–robot cooperation, where free-form human communication frequently exceeds the capability of protocol-bound robot agents Wan et al. (2022); Mandi et al. (2024). To address these issues, recent research has explored LLM-powered agents that communicate through natural language, with the Cooperative Embodied Language Agent (CoELA) framework serving as a representative example Zhang et al. (2024).

CoELA advances multi-agent interaction by enabling free-form dialogue, but its decision-making largely focuses on immediate, single-step actions without maintaining structured long-term plans. In contrast, Cooperative Plan Optimization (CaPo) Liu et al. (2025) introduces meta-planning to enhance coordination, but relies on rigid, rule-based communication protocols. In such protocols, robots communicate by following predefined discussion rules or debate-like turn-taking structures, which ensures precise progress tracking in robot–robot settings Seo et al. (2025). However, this rigidity creates two practical limitations. First, repeated discussion sessions can introduce substantial communication overhead Li et al. (2025). Second, when interacting with humans who use free-form and informal language, these strict rules often fail to support accurate and natural communication. Effective cooperation therefore requires agents that can infer goals from unconstrained dialogue, maintain consistent long-term strategies, and adapt plans dynamically without depending on strict communication rules.

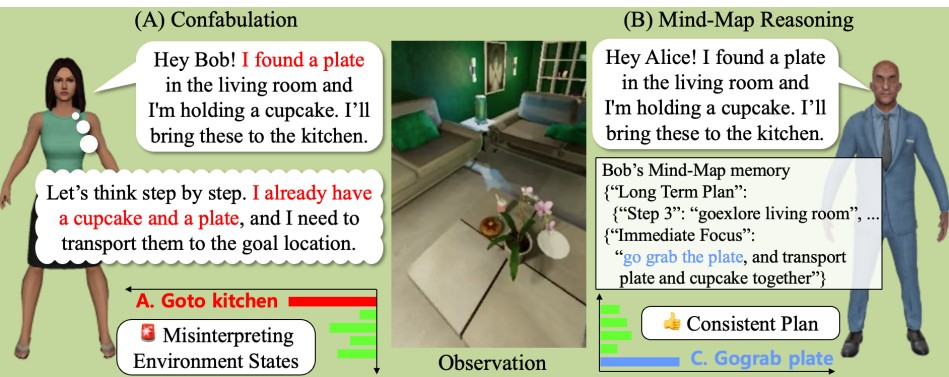

Figure 1: Comparison between confabulation and Mind-Map Reasoning in multi-agent communication. (A) Confabulation: The agent misinterprets dialogue as executed action, assuming possession of an object and leading to incomplete task execution. (B) Mind-Map Reasoning: By explicitly separating long-term plan and current state, the agent identifies missing objects and selects consistent actions, thereby mitigating confabulation and improving cooperation efficiency.

Another critical barrier emerges in hallucination and confabulation Smith et al. (2023); Sui et al. (2024), where LLMs may misinterpret dialogue history as factual state, leading to faulty inferences and redundant actions. This risk becomes more likely with unconstrained Chain of Thought (CoT) prompting Wei et al. (2022), where an agent may conflate spoken intentions with executed actions or assume environmental changes that have not actually occurred. For instance, as shown in Figure 1(A), an agent might treat a previous utterance as if the action had already been completed, disrupting task execution. In comparison, Figure 1(B) illustrates how a structured Mind-Map representation that separates dialogue, current state, and long-term plan can support more consistent decision-making, which in turn may reduce unnecessary retries and communication overhead Zhang et al. (2024); Guo et al. (2024).

To address these limitations, we introduce the Mind-Map Agent, an LLM-powered collaborative agent that integrates long-term planning with structured reasoning. At the core of our approach is Mind-Map reasoning, which maintains explicit representations of teammates' intentions, cooperative strategies, and communication plans. This structured representation helps agents remain consistent across interactions, mitigating confabulation and reducing redundant dialogue. By jointly capturing conversational cues and factual states, the approach is also applicable to human–robot collaboration, where agents must flexibly interpret free-form instructions without relying on rigid communication protocols.

- We introduce the Mind-Map Agent, which equips each agent with structured long-term reasoning by explicitly separating dialogue, task state, and cooperative strategies. This design enhances consistency in multi-step execution while reducing confabulation and redundant communication.

- We conduct extensive evaluation across diverse open-source LLMs, showing that Mind-Map reasoning consistently outperforms CoELA. These results demonstrate that structured reasoning provides robustness across model scales and architectures.

- Our analysis highlights both the benefits and limitations of Mind-Map reasoning. While it substantially mitigates confabulation, lightweight models remain more vulnerable in visual observation settings, underscoring ongoing challenges for current LLMs in generating reliable, grounded dialogue in collaborative tasks.

## 2 RELATED WORK

### 2.1 MULTI-AGENT COOPERATION

Research on multi-agent cooperation has developed a wide range of methods for coordination, communication, and decision-making in decentralized environments Stone & Veloso (2000). Early ap-

proaches such as MADDPG Lowe et al. (2017) introduced reinforcement learning techniques to improve strategic coordination among agents in cooperative–competitive settings Shu & Tian (2019). Subsequent work explored cooperation in embodied household tasks Jain et al. (2020), communication efficiency Liu et al. (2025); Patel et al. (2021), and dialogue grounding in environmental contexts Jain et al. (2019). Despite these advances, many methods still rely on opaque message-passing channels such as continuous vector embeddings Jain et al. (2019), which limit interpretability and adaptability across heterogeneous agents. Moreover, protocol-driven communication and centralized planning, long established in robotic systems with networked control, scale poorly to open human-facing environments. For multi-agent systems to collaborate effectively with humans, agents must make autonomous decisions under partial observability Spaan et al. (2006); Chang et al. (2025), coordinating tasks without globally shared knowledge or fixed communication protocols.

## 2.2 Cooperation with Natural Language Communication

Natural language has emerged as a promising alternative for enhancing cooperation, task planning, and information sharing in embodied multi-agent systems. Frameworks such as CoELA Zhang et al. (2024) integrate large language models (LLMs) to support flexible dialogue between agents and humans, enabling partners to share intentions and adapt cooperative behaviors. Built on the DEC-POMDP framework Bernstein et al. (2002); Goldman & Zilberstein (2003) and echoing cognitive architectures such as Soar Laird (2022), CoELA organizes decision-making into perception, memory, communication, planning, and execution modules. While these systems benefit from the reasoning capabilities of LLMs, challenges arise when models trained on general dialogue corpora are used for situated collaboration. Agents may generate irrelevant or emotional messages (e.g., utterances with exaggerated punctuation or emoji-like expressions), hallucinate actions such as declaring that an object has been grasped before execution, or even confuse roles by redundantly performing tasks already delegated to teammates. Such forms of confabulation occur when generated text is mistaken for factual observation or when dialogue history is misinterpreted as action history, leading to failures in planning and coordination. These limitations highlight the need for methods that constrain utterances to task-relevant content and incorporate structured reasoning to ensure reliable cooperation.

## 2.3 Optimizing Cooperation and Communication Strategy

Beyond enabling natural language communication, another line of work has focused on optimizing the balance between communication frequency and cooperative efficiency Liu et al. (2025); Li et al. (2025); Seo et al. (2025). For instance, CaPo Liu et al. (2025) introduces meta-planning to generate shared high-level strategies and adjust execution adaptively, while REVECA Seo et al. (2025) constrains dialogue through structured templates to support mutual awareness. While effective in controlled simulations, these frameworks assume rigid protocols such as always-on channels, explicit turn-taking, or repeated deliberation rounds. Such mechanisms provide transparency in robot–robot cooperation but introduce significant overhead Li et al. (2025) and reduce adaptability in human–robot interaction, where free-form dialogue is the norm. In practice, repeated meta-planning debates and verbose exchanges risk overwhelming both agents and human partners, especially when lightweight LLMs amplify "chattering" behavior that contaminates dialogue context and increases confabulation. These challenges highlight the need for mechanisms that support reliable, interpretable, and efficient communication without rigid protocol assumptions.

## 3 Mind Generation for Cooperative Planning

### 3.1 Mind-Map Agent: System Architecture and Operating Principle

Figure 2 presents the overall architecture of the Mind-Map Agent. The system is composed of modular components spanning from the internal agent modules to the external task environment. At its core, the agent operates with two distinct phases of large language model (LLM) reasoning: (1) Mind-Map Generation and (2) Plan Selection. Both phases make use of two complementary memory systems. The Text Interface functions as the agent's working memory, while the Mind-Map Memory provides a structured long-term memory that is explicitly maintained by the agent.

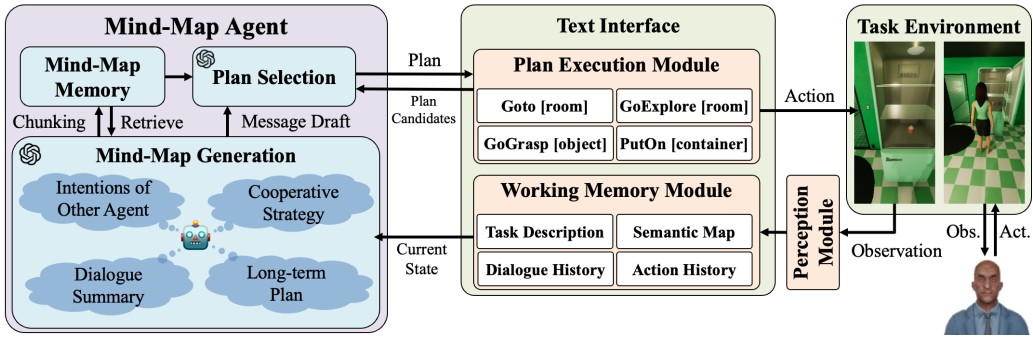

Figure 2: Overall architecture of the Mind-Map Agent. The system integrates two phases of LLM-based reasoning (Mind-Map Generation and Plan Selection) with complementary memory systems (Text Interface as working memory and Mind-Map Memory as structured long-term memory). The architecture supports perception-action loops while maintaining structured knowledge for long-horizon cooperative planning.

The Mind-Map Generation phase constitutes the first reasoning step. In this step, the LLM takes the current state from the Text Interface and integrates it with the stored Mind-Map Memory. The output is an updated structured mind map that may include interpretations of intentions, cooperative strategies, dialogue summaries, and long-term plans. This process combines incoming observations with previously stored task knowledge. By refreshing the Mind-Map Memory over time, the agent can maintain a stable representation that persists across interactions. This design contributes to a clearer separation between short-term environmental descriptions and the structured long-term task context, which may help reduce confusion between immediate perception and accumulated memory.

The Plan Selection phase forms the second reasoning step. In this step, the LLM references both the updated Mind-Map Memory and the Text Interface. Based on this combined context, the model determines the next action or communication. Outputs are passed to the Plan Execution Module, which translates high-level plans into primitive operations such as navigation or object manipulation. Alternatively, the Plan Selection may generate a dialogue message to be transmitted to teammates. The reliance on both working memory and long-term structured memory is intended to keep each decision connected to the immediate environment while also aligned with the broader cooperative plan.

The Text Interface serves as the working memory of the agent and includes four elements: task description, dialogue history, semantic map, and action history. This interface reflects what the agent can directly access at runtime and interacts with the task environment through the Perception Module and the Plan Execution Module. The Perception Module converts raw environment signals into symbolic observations that are stored in the Text Interface. The Plan Execution Module translates symbolic action plans into concrete operations that affect the environment and generate new observations, thereby closing the perception-action loop. In contrast, the Mind-Map Memory maintains a structured representation that is chunked and retrieved explicitly through the LLM. This long-term memory captures higher-level constructs such as inferred intentions, cooperative strategies, and reasoning outcomes. The distinction between these two memory systems is important: the working memory is transient and environment-driven, whereas the long-term memory is agent-managed and schema-based.

This dual-phase architecture draws on insights from cognitive architectures and multi-agent cooperation research, which emphasize that effective collaboration depends on the ability to infer and represent the hidden goals of others, often described as Theory of Mind (ToM) Lim et al. (2020); Wu et al. (2020); Ying et al. (2024). Building on these principles, the Mind-Map Agent transforms implicit intention inference into explicit structured entries that can be transparently maintained and reasoned over. This design contrasts with prior frameworks such as CoELA Zhang et al. (2024), which pioneered natural language based coordination in multi-agent systems but primarily relied on step-by-step reactive reasoning without durable long-term structures. Inspired by recent advances that frame LLMs as both reasoning agents and generators of structured knowledge Xu et al. (2024), as well as work on multi-persona collaboration for emergent reasoning Wang et al. (2023), the Mind-Map Agent introduces structured memory to clearly separate dialogue history, environmental state,

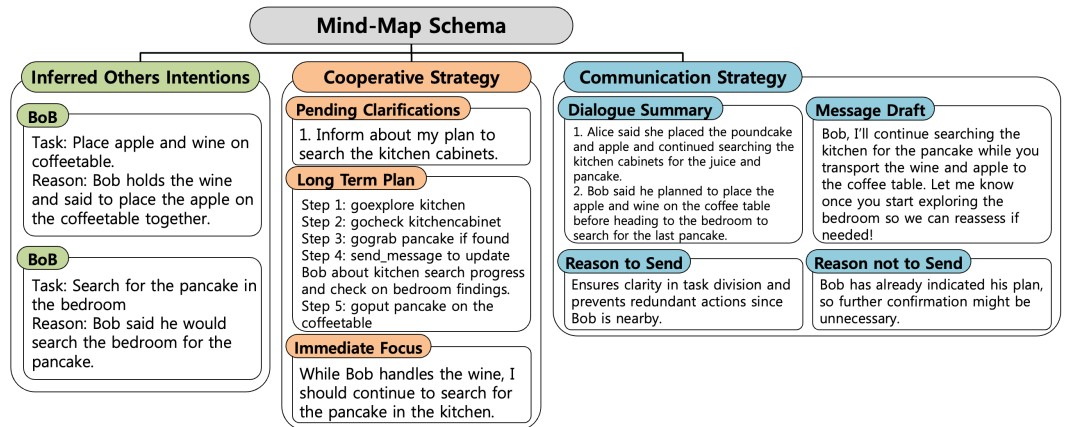

Figure 3: Mind-Map schema as maintained by the agent. The schema is divided into three main components: inferred others' intentions, cooperative strategy, and communication strategy. Together, they provide a structured representation of team state and guide reasoning over multi-agent and human–robot collaboration.

and cooperative strategy. This explicit representation reduces redundant or ambiguous utterances and enables reliable tracking of completed versus pending actions, thereby supporting more robust long-horizon cooperation in both multi-agent and human–robot contexts.

## 3.2 MIND-MAP SCHEMA AND KEY COMPONENTS

At the core of our approach is the Mind-Map schema, a structured representation of the agent's understanding of team state and task progress. During the Mind-Map Generation phase, the agent is guided by in-context exemplars expressed in a JSON format, ensuring that updates remain structured and consistent. The schema is dynamically updated as new observations and utterances are received. As illustrated in Figure 3, the schema consists of three main components: inferred others' intentions, cooperative strategy, and communication strategy. Together, they enable the agent to track teammates' goals, maintain alignment with the joint plan, and regulate communication to avoid redundant or irrelevant utterances.

**Inferred Others' Intentions** This component guides the agent to interpret teammates' utterances as explicit expressions of intent and transform them into structured entries within the mind map. In this way, natural language is not only treated as surface-level communication but also as a representation of hidden goals and action plans. By recording and organizing these inferred intentions, the agent externalizes a Theory-of-Mind perspective Lim et al. (2020); Wu et al. (2020), enabling it to track what others aim to achieve and why. This structured account of intent can be reused in subsequent reasoning steps, allowing the agent to anticipate teammates' actions, align its own behavior with theirs, and reduce the need for repeated clarification. In this way, the process provides a systematic mechanism for capturing language-mediated intentions, supporting consistent interpretation of teammates' goals and facilitating alignment in cooperative decision-making.

**Cooperative Strategy** This component organizes the agent's reasoning about how to align its own role with the collective plan. It is structured into three layers: Pending Clarifications, which track unresolved issues requiring coordination with teammates; a Long-Term Plan, which outlines the global sequence of steps needed for successful task completion; and an Immediate Focus, which identifies the agent's current local priority. These layers ensure that the agent's decisions remain consistent across different time horizons. Pending Clarifications reduce ambiguity in teamwork by explicitly marking information gaps, while Long-Term Plans help prevent duplicated efforts through clear division of labor. Immediate Focus links ongoing execution to higher-level goals, reducing the risk of short-sighted or conflicting proposals.

**Communication Strategy** This component frames the agent's process of generating utterances by embedding them in the dialogue context and requiring explicit justification before transmission. It is

divided into four sub-elements: Dialogue Summary, which condenses prior utterances into a concise history to prevent redundancy; Message Draft, which formulates candidate utterances grounded in the cooperative plan; and explicit reasoning fields that capture the Reason to Send and Reason not to Send. This reasoning process is integrated with the subsequent action selection phase, ensuring that the necessity of communication is evaluated before a message is produced. By discouraging unnecessary utterances, the Communication Strategy reduces coordination overhead and ensures that each message serves a clear collaborative function.

Building on these components, the Mind-Map Agent maintains a structured representation that guides foresight and long-horizon collaboration. Each new observation or teammate utterance updates the relevant fields of the schema, which in turn informs both action selection and communication decisions. By integrating these structured reasoning layers, the agent gains the ability to plan over longer horizons compared to single-step reactive approaches Zhang et al. (2024), while reducing coordination overhead that often arises in meta-plan deliberations Liu et al. (2025). In contrast to goal-alignment methods that emphasize convergence on a shared overall objective Ying et al. (2024), the Mind-Map Agent complements this perspective by highlighting explicit and distributed articulation of sub-goals within the shared mission. To maintain coherence, updates are constrained to a JSON schema and guided by in-context exemplars, which reduces hallucinations caused by ambiguous or off-topic utterances. The schema thus functions as an externalized working memory that can be inspected to reveal the agent's reasoning process, thereby supporting transparency and interpretability in multi-agent and human–robot cooperation.

## 4 EXPERIMENTS

### 4.1 IMPLEMENTATION AND BASELINES

**Benchmarks.** We evaluate our approach on two cooperative embodied AI benchmarks from the CoELA framework. The first is Communicative Watch-and-Help (C-WAH) Puig et al. (2020; 2018), which focuses on household tasks that require two agents to coordinate through natural language communication. The second is ThreeDWorld Multi-Agent Transport (TDW-MAT) Gan et al. (2022), where agents operate in a physically realistic environment to transport objects using containers. While C-WAH emphasizes communication and joint planning, TDW-MAT evaluates efficiency in collaborative object delivery.

**Baselines.** We compare the proposed Mind-Map Agent against both classical planning-based agents and LLM-based agents. The classical baselines include a Monte Carlo Tree Search hierarchical planner (MHP) and a rule-based hierarchical planner (RHP), both of which execute pre-defined routines without language use. The single-agent MHP setting serves as a reference for evaluating efficiency, while multi-agent MHP and RHP provide stronger non-LLM baselines. On the LLM side, we adopt the CoELA agent architecture and vary the LLMs: GPT-4o-mini, GPT-OSS-120B, Qwen-7B, and Llama3.3-70B. For each models, we evaluate both the CoELA variant and our Mind-Map variant to assess the contribution of structured reasoning. In addition, we include a Random Agent baseline which without LLM and selects randomly among plan candidates proposed by the Plan Execution Module. The Random Agent provides a lower bound and highlights the net contribution of LLM reasoning.

**Metrics.** For C-WAH, we measure cooperative efficiency using Average Steps (AS), defined as the number of steps required to complete all target tasks, and communication efficiency using Average Communication Steps (ACS), defined as the average number of utterances per episode. Lower values in both metrics indicate higher efficiency. To ensure robust evaluation, we repeat each experiment three times and report the mean and standard deviation. For TDW-MAT, we use Transport Rate (TR), defined as the fraction of target objects successfully delivered within 3000 timesteps, where higher values indicate better performance.

### 4.2 EXPERIMENTAL RESULTS

#### 4.2.1 COMMUNICATIVE WATCH-AND-HELP

The results on C-WAH, summarized in Table 1, indicate that the Mind-Map Agent tends to outperform both classical planners and other LLM-based agents in terms of task completion efficiency

Table 1: C-WAH performance across different agents. All values are reported as mean ± standard deviation across three runs. Lower is better (↓).

| Model Type | | Classic Agent MHP | | GPT-4o-mini | | GPT-OSS-120B | |
|---|---|---|---|---|---|---|---|
| | | Single | Multi | CoELA | MindMap | CoELA | MindMap |
| Symbolic Obs | AS↓ | 111 | 75 | 65.4±23 | 57.0±19 | 74.3±21 | 53.2±18 |
| | ACS↓ | - | - | 12.1±5.0 | 1.5±1.5 | 4.2±2.0 | 4.3±3.0 |
| Visual Obs | AS↓ | 141 | 103 | 113.9±37 | 98.1±18 | 125.2±40 | 94.0±27 |
| | ACS↓ | - | - | 15.5±6.2 | 2.1±1.7 | 3.9±2.0 | 5.5±4.1 |
| Model Type | | Random Agent | | Qwen2.5 7B | | Llama3.3 70B | |
| | | Single | Multi | CoELA | MindMap | CoELA | MindMap |
| Symbolic Obs | AS↓ | 148±33 | 91.9±21 | 90.4±27 | 72.6±21 | 71.2±19 | **53.1±17** |
| | ACS↓ | - | 0 | 12.1±3.3 | 9.8±4.5 | 4.1±2.0 | **0.9±1.0** |
| Visual Obs | AS↓ | 186±34 | 128±29 | 174.1±25 | 134.3±10 | 128.6±39 | **90.6±25** |
| | ACS↓ | - | 0 | 14.9±21 | 10.2±5.1 | 5.1±2.6 | **1.3±1.5** |

and communication cost. Beyond the quantitative outcomes, several qualitative observations can be drawn.

First, smaller LLMs such as Qwen-7B often perform less effectively than even the non-reasoning Random Agent. This suggests that unstructured or hallucinated dialogue may in some cases interfere with coordination, resulting in less efficient task execution. For instance, we observed that Qwen2.5-7B frequently produced conversational turns containing emojis or multi-turn self-dialogues, reflecting tendencies from its pretraining data. Such behaviors, while natural in casual dialogue, were less suitable for cooperative task execution in this benchmark.

Second, the Mind-Map representation appears to mitigate these issues by separating dialogue history from the factual task state. This structured reasoning helps smaller models generate more consistent plans and, in some cases, allows them to approach or surpass the performance of much larger models without structured support. Explicit schema guidance focuses the agent on relevant task elements, leading to plans that are more coherent and aligned with the environment. It is also notable that, across most models, the CoELA baseline shows performance levels comparable to the Random Agent, indicating that language reasoning alone does not necessarily improve efficiency without structured memory. In part, this may be due to the CoELA prompt design being optimized for GPT-4, which other open models did not always follow strictly, resulting in mismatched or incomplete action selection.

Finally, Mind-Map Agents generate fewer utterances in most cases while maintaining effective cooperation. For example, with stronger LLMs such as Llama3.3-70B, the Mind-Map Agent achieves near-silent coordination once successful inference of the partner's intentions has been established. In these cases, the agent appears to capture teammates' goals with greater reliability and, based on environmental observations alone, can often construct long-horizon plans while monitoring the state of other agents. This enables effective collaboration with minimal reliance on explicit communication. The results suggest that structured reasoning not only facilitates early alignment of plans but also supports adaptation to evolving team states. By reducing the need for repeated exchanges and redundant clarifications, the Mind-Map Agent lowers overall communication overhead while sustaining cooperative efficiency in complex multi-agent settings.

### 4.2.2 THREEDWORLD MULTI-AGENT TRANSPORT

The advantages of the Mind-Map approach also extend to the TDW-MAT transport challenge (Table 2). In this setting, Mind-Map agents consistently achieve higher transport success rates than the baselines by dividing labor more effectively and by leveraging containers to reduce redundant trips. Structured reasoning enables agents to exploit environment dynamics more systematically. While CoELA agents often rely on single-object deliveries, Mind-Map Agents employ containers to transport multiple objects at once, leading to faster progress and higher success rates within the timestep limit. Beyond task division, the Mind-Map approach also supports efficiency when agents

Table 2: TDW-MAT performance across different agents.

| Transport Rate ↑ | Classic Agent RHP | | GPT-4o-mini | | Llama3.3 70B | |
|---|---|---|---|---|---|---|
| | Single | Multi | CoELA | MindMap | CoELA | MindMap |
| Food | 52% | 76% | 65% | 86% | 76% | **86%** |
| Stuff | 49% | 74% | 76% | 79% | 74% | **83%** |
| Total | 50% | 75% | 71% | 82% | 75% | **85%** |

act independently. An agent may, for example, defer object retrieval until locating a container, or coordinate with a teammate regarding container usage. When one teammate is already carrying two objects, the other can provide information about the location of a container in a different room, thereby facilitating cooperation. These behaviors indicate that explicit schemas of intentions and strategies contribute to more stable teamwork, suggesting that the Mind-Map approach constitutes a generalizable mechanism for enhancing decentralized multi-agent collaboration.

## 4.3 ABLATION STUDY ON IN-CONTEXT EXAMPLE OF THE MIND-MAP COMPONENT

To examine which elements of the mind map contribute most to performance, we conduct an ablation study on C-WAH using GPT-4o-mini model. The agent is prompted to generate different subsets of the mind map, and performance is evaluated using Average Steps (AS) and Average Communication Steps (ACS). The tested settings include *Inf* (Inferred Others' Intentions) only, *Coop* (Cooperative Strategy) only, *Comm* (Communication Strategy) only, and their combinations (Inf+Coop, Inf+Comm, Coop+Comm, and Inf+Coop+Comm). For this study, the key-matching process was deliberately omitted, allowing the agent to freely generate components without explicit structural constraints.

Table 3: Ablation study on different subsets of the Mind-Map component in C-WAH using GPT-4o-mini. Average Steps (AS), Average Communication Steps (ACS)

| Mind-Map Example | AS ↓ | ACS ↓ |
|---|---|---|
| Inf | 66.6 | 6.0 |
| Coop | 59.6 | 6.5 |
| Comm | 51.5 | **1.6** |
| Inf + Coop | **49.4** | 4.8 |
| Inf + Comm | 60.2 | 2.9 |
| Coop + Comm | 62.9 | 3.3 |
| Inf + Coop + Comm | 55.6 | 2.4 |

First, intention inference alone appears insufficient for effective collaboration. When relying only on inferred intentions (Inf), the agent exhibits higher step counts and substantial dialogue compared to the full Mind-Map agent. This suggests that predicting partner intentions without cooperative or communication planning does not provide enough structure for efficient execution.

Second, combining Cooperative Strategy and Communication Strategy (Coop+Comm) without explicit intention inference results in frequent overlapping behaviors. In this case, both agents often produced similar sub-plans independently, indicating that intention modeling is essential for task division and avoiding redundant effort.

Third, the Inf+Coop combination achieves the best numerical performance, with lower step counts than other subsets. However, in this setting, the agent did not generate the intended *Message Draft* within the mind map, instead defaulting to CoELA's message generation. This observation suggests that refining the prompting process for Communication Strategy, perhaps with multi-phase message drafting, may further improve coordination when all components are included.

Finally, in the Comm-only setting, the agent displayed emergent flexibility by autonomously introducing additional elements beyond the predefined structure. These included tentative action plans, collaboration tactics, and shared goals, indicating that the agent adapts the reasoning structure when provided with minimal guidance. Such behavior highlights the potential value of allowing flexible mind map generation, where the model supplements missing components to support effective cooperation.

Overall, the ablation results suggest that diverse in-context examples of structured reasoning help agents organize their decision-making. Each component plays a distinct role: intention inference reduces redundancy, cooperative strategy supports task division, and communication strategy streamlines dialogue. When used together, they provide a balanced framework for adaptive and efficient multi-agent teamwork. Moreover, this result reveals that agents can correct and update their reasoning across steps, effectively managing their own long-term memory to support sustained cooperation.

## 5 CONCLUSION

### 5.1 LIMITATIONS AND DISCUSSION

While the experimental results demonstrate clear advantages of the Mind-Map Agent, several limitations warrant consideration. First, our evaluation is confined to two simulation-based benchmarks that include intentionally unrealistic states (e.g., a fork placed inside a refrigerator). These settings are useful for assessing coordination efficiency in tasks with high dialogue requirements, but they also restrict the scope of validation. Second, smaller LLMs remain unstable and frequently produce noisy utterances. Their performance is highly sensitive to prompt design and schema exemplars, raising concerns about reproducibility across different model architectures. Third, the Mind-Map reduces confabulation and lowers the number of model calls but introduces additional prompt overhead. This trade-off reflects a shift from efficiency to cautious reasoning, a compromise appropriate in safety-critical domains. Finally, the robustness of Mind-Map reasoning has so far only been examined in simulation-based settings. Future research should include evaluations with real human collaborators to validate the framework in interactive and open-ended cooperative scenarios.

Despite these limitations, several important insights emerge from our study. Mind-Map reasoning substantially mitigates confabulation by explicitly separating dialogue, factual state, and cooperative strategies, thereby improving consistency in multi-agent communication. Moreover, emergent behaviors were observed in the Comm-only setting, where the agent autonomously expanded the reasoning structure to incorporate missing elements such as collaboration tactics or shared goals. This suggests that schema-guided yet flexible reasoning enables agents to compensate adaptively when the provided structure is incomplete.

Another promising aspect lies in the transparency afforded by the Mind-Map. By externalizing hidden goals and strategies into a structured memory, agents produce interpretable representations that can facilitate human–AI collaboration. Nevertheless, a trade-off arises: reducing explicit communication can improve efficiency but may simultaneously reduce transparency for human teammates. Future research should therefore investigate how to balance communication efficiency with interpretability, particularly in collaborative scenarios that require mutual trust and shared situational awareness.

### 5.2 CONCLUSION

This work introduced the Mind-Map Agent, a framework for structured reasoning in multi-agent cooperation. By disentangling dialogue, task state, and cooperative strategies, the approach mitigates confabulation, preserves long-term consistency, and improves interpretability. Empirical evaluation across multiple LLMs and environments demonstrated consistent gains over both classical planners and unstructured LLM-based agents, underscoring the complementary value of intention inference, cooperative strategy, and communication strategy.

Beyond quantitative improvements, the findings highlight a broader implication: structured reasoning and memory offer a principled means of reconciling rigid, protocol-driven planning with unconstrained, dialogue-based coordination. Although our study was conducted in simulation-based benchmarks, the findings suggest directions for extending reliable multi-agent interaction to real-world scenarios, including human–robot collaboration.

While open challenges remain in grounding perception, scaling to lightweight models, and adapting to open-ended human interaction, the proposed framework contributes toward the development of multi-agent systems that are not only more robust and efficient but also more transparent and better aligned with real-world cooperative demands.

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
