# A    THEORETICAL FRAMING OF DECENTRALIZED EMBODIED COOPERATION

## A.1    DECENTRALIZED CONTROL UNDER DEC-POMDP

Decentralized embodied multi-agent cooperation is formally captured by the Decentralized Partially Observable Markov Decision Process (Dec-POMDP) framework (Bernstein et al., 2002; Goldman & Zilberstein, 2003). In a Dec-POMDP, $N$ agents operate without a central controller, each making decisions based on its own local observations and policy. Formally, a Dec-POMDP can be defined by a tuple

$$(N, S, \{A_i\}_{i=1}^N, \{O_i\}_{i=1}^N, T, R, \gamma, h),$$

where $S$ is the state space, $A_i$ and $O_i$ are the action and observation spaces for agent $i$, $T$ is the state transition function, $R$ is a shared team reward function, $\gamma$ is a discount factor, and $h$ is the horizon. Each agent $i$ selects actions $a_i \in A_i$ based on its own observation history, without access to other agents' private observations or hidden state, while the team jointly maximizes the common reward. There is no centralized coordinator that broadcasts joint decisions; coordination must arise from the agents' local policies and interactions. As a result, Dec-POMDPs are significantly more challenging than single-agent or centrally controlled POMDPs, because each agent faces uncertainty both about the environment state and about its teammates' internal information (Bernstein et al., 2002).

Modern formulations often extend Dec-POMDPs to include *communication actions* that let agents share information at a cost (Goldman & Zilberstein, 2003; 2004). In such extensions, an agent's action space $A_i$ is partitioned into physical world actions and communication actions (for example, sending a message $\sigma_i$), and its observation space $O_i$ includes both world observations and any messages received from teammates. This yields a principled way to model multi-agent cooperation where communication is possible but not free. Even in this extended setting, control remains decentralized: each agent decides *when* and *what* to communicate based only on local information, rather than querying a central hub. This decentralized control formalism underpins the embodied multi-agent cooperation setting used in CoELA (Zhang et al., 2024) and in our Mind-Map Agents.

## A.2    INDIRECT COMMUNICATION VIA OBSERVATIONS

Classical work on decentralized control distinguishes several ways in which cooperative agents can share information during execution. In particular, Goldman & Zilberstein (2004) identify three mechanisms:

1. **Indirect communication via observations.**    Agents exchange information through environment-mediated signals: changes in object configuration, a teammate's flashlight, gestures, or spoken natural language that propagate through the world and appear in each agent's local observation stream.

2. **Direct communication via messages.** Agents send protocolized messages over a dedicated channel provided by the multi-agent system, such as shared memory or an internal message bus. These messages are not constrained by physical perception and are typically delivered synchronously and without noise.

3. **Common uncontrollable features.** Agents may observe shared state variables that are not influenced by their actions (e.g., weather), and can use these as implicit coordination signals.

In decentralized embodied systems, agents cannot assume an omniscient message-passing service, especially when collaborating with humans. All information exchange must arise through the environment or through explicit communication actions. We use the term *indirect communication via observations* (Goldman & Zilberstein, 2004) for any information transfer that is mediated by the environment rather than by a dedicated synchronized channel. One agent performs an action that changes the environment, and another agent infers information from its own observations. For example, an agent may point at a location, move an object into view, or utter a sentence so that its teammate can perceive the effect as part of its sensory stream.

From the Dec-POMDP/Dec-MDP-Com perspective (Goldman & Zilberstein, 2004), this style of interaction can be viewed as treating both physical actions and communicative acts as environment transitions that generate observations. CoELA explicitly instantiates this assumption in embodied

multi-agent settings: natural-language utterances are issued as actions in the environment and received as observations, and agents have no privileged access to each other's internal memories or task histories (Zhang et al., 2024). Our experiments inherit this communication model: messages are planned in the same space as other actions and consume the same step budget, rather than being handled by a separate oracle channel.

In our embodied setting, human–robot and multi-embodiment collaboration therefore relies on physical signals that propagate through the environment. A robot's message to a human partner is perceived through the human's own sensors as part of the world, not as a privileged side channel. Any communicative act must be realized as an environment action (for example, speaking a sentence or displaying a visual cue) that becomes part of the partner's sensory input. This view aligns with the decentralized settings studied in CoELA and related benchmarks, where agents operate under partial observability without access to a partner's internal task history or privileged simulator state (Zhang et al., 2024; Chang et al., 2025).

Our experimental setting follows this line. Agents can obtain information about teammates only through (i) partial visual observations and (ii) natural-language messages that count as environment steps and explicitly contribute to the step budget. We do not assume an independent, cost-free message channel that bypasses the environment. This stands in contrast to centralized or semi-centralized multi-agent LLM controllers such as CaPo (Liu et al., 2025), REVECA (Seo et al., 2025), and CoTS (Zu et al., 2025), where communication is implemented as protocolized intra-system messages with guaranteed delivery and does not count as interaction with the world. In our multi-embodiment regime, all information sharing is effectively *indirect communication via observations*, even when it takes the form of natural-language dialogue.

### A.3 CENTRALIZED AND SEMI-CENTRALIZED ARCHITECTURES IN PRIOR WORK

Several recent multi-agent LLM frameworks deviate from the strictly decentralized paradigm by introducing centralized or semi-centralized forms of coordination. We briefly analyze CaPo, REVECA, and CoTS, and contrast them with fully decentralized methods such as CoELA and our Mind-Map Agents.

**CaPo (Cooperative Plan Optimization).** Liu et al. (2025) adopts a *semi-centralized* strategy through structured multi-turn communication. In the first phase, the agents engage in a turn-taking dialogue to form a shared meta-plan. One agent is designated to take the lead in plan synthesis, aggregates information from others, and proposes a complete task decomposition that allocates subtasks to each agent. The remaining agents then provide feedback or corrections. This process repeats in synchronized discussion rounds until consensus is reached or a communication budget is exhausted. The protocol assigns a special role to the *plan proposer* and expects all agents to respond according to the dialogue schedule, which is effectively a forced-response protocol.

Communication in CaPo thus follows a global schedule. Each round of meta-planning assumes reliable messaging and participation from all agents. This design promotes coherent and long-horizon joint plans, but it also presupposes that agents can be paused and synchronized for discussion. In practice, CaPo constructs a shared plan in a centralized manner through the reasoning of a single meta-planner before decentralized execution begins. This is closely related to the "centralized training with decentralized execution" (CTDE) paradigm in multi-agent reinforcement learning, as made explicit in LIET (Li et al., 2025). Applying this scheme to heterogeneous robots or human partners would require strong assumptions about connectivity and responsiveness, and any failure in the protocol could undermine the planning process.

**REVECA (Relevance and Validation-Enhanced Cooperative Agent).** REVECA is built on the CoELA architecture and uses LLMs to control each agent under partial observability, with communication through natural language as in CoELA (Seo et al., 2025). On top of this, REVECA introduces additional modules for relevance-based memory management and a Validation Module that checks proposed plans against collaborators' potential actions and trajectories. To perform this validation, REVECA retrieves collaborator-related information from memory and dialogue logs, and reasons about where the teammate may have moved and which objects may already have been handled.

This design improves coordination efficiency by reducing false plans, but it also moves the system away from strictly local reasoning. The validation step requires each agent to reason over a joint picture of team progress that is constructed from the collaborator's action history and position, rather than relying solely on its own observations or sparse messages. Moreover, when REVECA decides that confirmation is needed, it sends explicit messages that are assumed to be delivered and answered by the partner's communication module. In the simulator, this implements a reliable direct channel: a message is injected into the other agent's memory, and the sending agent directly invokes the partner's response function. In this sense, REVECA remains decentralized at the level of separate execution processes, but introduces a form of shared oversight through systematic retrieval of collaborator memory and through guaranteed message responses.

**CoTS (Collaborative Tree Search).** CoTS introduces a global planning procedure in which multiple agents reason within a single shared search tree (Zu et al., 2025). A modified Monte Carlo Tree Search (MCTS) algorithm constructs and evaluates joint action sequences for all agents, guided by an LLM-based reward model. The tree nodes encode candidate future joint plans, including both actions and dialogue. At decision time, CoTS selects a joint plan from this tree and then dispatches the corresponding actions or messages to each agent.

In effect, CoTS treats the multi-agent team as a composite planner. The central search tree synchronizes decision-making, and the key planning steps occur in a single global reasoning process. Communication among agents is not emergent, but is orchestrated by the tree search: the algorithm decides when agents "talk," which suggestions are exchanged, and when the plan should be revised. All messages and plans live in a shared data structure, and agents execute what the centralized planner chooses. This assumption of a shared reasoning space with synchronized access is natural in a digital multi-agent system running in a single process, but it is difficult to realize for physically distributed robots without a central controller.

**Summary.** Across these examples, CaPo, REVECA, and CoTS each introduce centralized or semi-centralized assumptions: a designated meta-planner and synchronous plan refinement in CaPo; validation modules that reason over collaborator trajectories and guaranteed direct messages in REVECA; and a shared tree search that plans for all agents simultaneously in CoTS. These approaches often also assume that certain information, such as a meta-plan or search tree node, is available to all agents as a form of shared memory. Communication is treated as an internal operation of the algorithm rather than as a costly environment action that may fail or be delayed.

This stands in contrast to a strictly decentralized Dec-POMDP setting, where each agent must act on its own local history and any communication must be explicitly decided, executed, and observed. Centralized and semi-centralized methods can deliver strong performance in simulation by simplifying coordination, but they move the system toward a single-agent viewpoint in which one joint policy operates over global information.

## A.4 COMPARATIVE PERFORMANCE ON C-WAH BENCHMARK

To illustrate the practical impact of these modeling choices, we report performance on the Communicative Watch-And-Help (C-WAH) benchmark (Puig et al., 2020; Zhang et al., 2024), a long-horizon, multi-room household rearrangement task. Table 4 compares two centralized approaches (REVECA and CoTS) against two decentralized approaches (CoELA and our Mind-Map Agents) under symbolic observations.

Centralized systems (REVECA, CoTS) appear to complete tasks in fewer environment interaction steps ("Simulation Steps") than fully decentralized agents. However, this comparison ignores communication. In the centralized settings, agents use a direct messaging channel, reflected by the non-zero Direct Message Count. These direct messages do not consume simulation steps, because they are handled by an internal protocol that assumes instantaneous and reliable delivery. If we add the number of direct messages to the simulation steps ("Average Steps"), the effective interaction budget increases. For example, REVECA uses about 60.9 total steps when each direct message is counted as a step-equivalent, and CoTS rises from 60.8 environment steps to roughly 64.9 total steps.

Table 4: C-WAH symbolic observation performance across centralized and decentralized approaches. For centralized methods, direct messages are internal protocol messages that do not consume environment steps. For decentralized methods, communication is realized as explicit environment actions (indirect messages) that are counted as simulation steps.

| Model Type | Centralized | | Decentralized | |
|---|---|---|---|---|
| | REVECA | CoTS | CoELA | MindMap |
| Simulation Steps | 52.23±13.61 | 60.8±15.3 | 65.4±23 | 57.0±19 |
| Direct Message Count | 8.76±3.0 | 4.13±0.7 | 0 | 0 |
| Indirect Message Count | 0 | 0 | 12.1±5.0 | 1.5±1.5 |
| Average Steps | 60.9 | 64.9 | 65.4 | 57.0 |

By contrast, CoELA and Mind-Map Agents operate without such a free channel. Communication must be enacted through environment actions, which are counted in the Simulation Steps column as indirect messages. CoELA agents use about 12.1 environment actions that function as signals to the partner, whereas Mind-Map Agents need only about 1.5 such actions on average. Because communicative acts are already included in the step count, Simulation Steps and Average Steps coincide for decentralized models. The important point is that a "step" is not directly comparable across paradigms. Centralized teams may seem more efficient if communication is hidden, but they can be exchanging substantial information outside the step count. Decentralized teams must decide whether to spend steps on communication and pay that cost explicitly.

Mind-Map Agents achieve the lowest Average Steps among all four methods while using no direct channel. This shows that careful planning under Dec-POMDP constraints can match or exceed the efficiency of centralized approaches. The small number of indirect messages suggests that the Mind-Map Agent can often coordinate implicitly by maintaining richer internal models of the task and teammate, rather than relying on frequent signaling.

### A.5 DISCUSSION: IMPORTANCE OF CLARITY IN ASSUMPTIONS

These results underscore the importance of clearly stating communication and control assumptions in multi-embodiment agent research. If a method quietly relies on synchronous, guaranteed messages or shared memory, its reported performance may not transfer to settings where agents are truly independent, such as heterogeneous robot teams or human–robot collaboration without full connectivity. Conversely, a fully decentralized method may appear weaker if evaluation metrics do not account for the cost of communication in a consistent way.

As embodied AI moves toward real-world deployment, centralized and decentralized cooperation should be treated as different problem regimes, not as interchangeable details. Future work should specify whether a central controller exists, whether agents share memory, and whether communication is free or costly, and should relate these choices to Dec-POMDP-style formulations when appropriate (Bernstein et al., 2002; Goldman & Zilberstein, 2004). This clarity makes empirical comparisons meaningful and helps ensure that improvements in simulation correspond to gains in realistic multi-agent cooperation. Without it, there is a risk of conflating fundamentally different settings and obscuring the challenges that arise when agents must coordinate under partial observability, communication delays, and independent decision-making in the physical world.

## B EXAMPLES OF LLM-INDUCED CHATTERING IN COOPERATIVE EMBODIED TASKS

Despite the two-stage architecture of CoELA, comprising explicit message generation and subsequent plan selection, we empirically observe that lightweight LLM-based agents frequently engage in excessive, low-information exchanges, a phenomenon we refer to as *chattering*. Such chattering saturates the limited shared context window with redundant utterances, impedes concise state sharing, and ultimately degrades cooperative planning and execution efficiency.

To make this phenomenon concrete, we report representative dialogue fragments generated by CoELA in the C-WAH environment. In these traces, agents repeatedly restate already-established intentions, over-specify routine actions, and emit multiple near-duplicate messages within only a few time steps. These examples demonstrate how LLM-based agents can induce disproportionately verbose interaction patterns in multi-agent collaboration.

Each dialogue begins with the seed exchange:

> **Alice:** "Hi, I'll let you know if I find any goal objects and finish any subgoals, and ask for your help when necessary."
> **Bob:** "Thanks! I'll let you know if I find any goal objects and finish any subgoals, and ask for your help when necessary."

### B.1 SHORT-HORIZON REPETITION AND UNCERTAINTY

The most basic form of chattering manifests as **High-Frequency Lexical Repetition**. Here, the agent emits multiple variations of the same plan or request in rapid succession. As seen in **Example B.1**, Bob generates two nearly identical messages about checking the kitchen and bedroom. This behavior likely stems from the model's uncertainty regarding whether the first message was sufficiently clear or "committed" to the context.

---

**Example B.1: High-Frequency Lexical Repetition over Short Horizons**

**Bob:** Can you check the kitchen? I think the pudding and juice might be there. Also, maybe grab an apple while you're at it. I'll head to the bedroom to see if I can find the cupcakes. Let's meet back here to compare notes.

**Bob:** Can you check the kitchen for pudding, juice, and an apple? I'll go to the bedroom to look for cupcakes. Let's meet back here to compare notes.

---

**Alice:** Let's meet at the <kitchentable> (72) in the kitchen. I'll bring the cupcake and juice. Check the fridge or cabinets for wine and apple. Meet at the table!

**Alice:** Let's meet at the <kitchentable> (72). I'll bring the cupcake and juice. You check the fridge or cabinets for wine and apple. Meet at the table!

---

### B.2 AGREEMENT LOOPS AND ECHOING

Even when a plan is successfully agreed upon, agents often fall into **Mutual Echoing**. In **Example B.3**, the agents agree to meet at the coffee table but fail to proceed to the action phase. Instead, they enter an "agreement loop," continuously restating the meeting location.

---

**Example B.2: Mutual Echoing Around a Shared Subgoal**

---

**Alice:** "I'm in the bathroom and haven't found anything related yet. Have you explored the livingroom or kitchen? It would be great if you could check those areas for the wine, cupcakes, and pudding."

**Bob:** Can you check the living room or the kitchen? I haven't explored those areas yet. If you find any of the items, could you bring them to the coffeetable (247)? Thanks!

**Bob:** Also, I'm currently in the living room. If you're nearby, we could team up to speed things up.

**Alice:** Got it, I'll check the kitchen first since I'm already nearby. If I find the wine, cupcakes, or pudding, I'll bring them to the coffeetable. See you there!

**Bob:** Let's team up in the kitchen. I have the pudding and can grab the wine and cupcakes with you.

**Alice:** Let's team up in the kitchen. You take the wine and cupcakes while I get the pudding. We can meet by the coffeetable (247) to finish this task quickly.

**Bob:** Alice, I'm in the bathroom and didn't find anything useful. Let's head back to the kitchen to grab the remaining cupcake and meet by the coffeetable (247) to finish.

**Alice:** Let's head to the kitchen, I'm holding the cupcake. We can grab the wine from the kitchencabinet (136) or (137) and meet by the coffeetable (247) to finish.

**Bob:** Let's grab the remaining cupcake from the kitchen and head to the coffeetable (247) together to finish.

**Alice:** Let's grab the wine from the kitchencabinets (136 or 137) and meet by the coffeetable (247) to finish.

**Bob:** Let's go get the last cupcake and meet by the coffeetable (247) to finish.

**Bob:** Let's grab the last cupcake from the kitchen and head to the coffeetable (247) together to finish.

---

B.3    CONVERGENCE LOOPS AND CONTEXTUAL BLINDNESS

At the final stage of a task, agents may enter a **Convergence Loop**. **Example B.4** demonstrates this with the fridge interaction. Notably, the typo `rigerator` (from *refrigerator*) creates a unique token pattern that the agents blindly copy-paste in subsequent turns. This confirms that the agents are attending to the surface form of the dialogue history rather than the underlying state.

---

**Example B.3: Convergence Loops at the Final Goal Location**

---

**Bob:** Found 1 <juice> (381) and 1 <pudding> (383) in the fridge. I'm heading to the <dining room> (111) to look for 1 <cupcake> and 2 <apples>. Can you check the kitchen cabinets for the remaining items?

**Bob:** Can you check the kitchen cabinets for 2 more <apple>s (384, 385)? I'm in the dining room looking for 1 <cupcake>.

**Alice:** Bob, I found 2 apples in the kitchen. I'm in the kitchen with <fridge> (165) and some other items. You can bring the cupcake over when you find it. Let's work together to speed up!

**Bob:** Alice, got <cupcake> (382). Let's both head to the <fridge> (165) and place the items inside!

**Alice:** Let's move quickly! I have <juice> (381) and <pudding> (383), you have the <cupcake> (382). Let's combine forces and put everything in the fridge together! rigerator

**Bob:** Alice, on my way to the kitchen with <cupcake> (382) and the <juice> (381). Let's put them all in the fridge together! rigerator

**Bob:** Alice, heading to the kitchen with <cupcake> (382) and <apple> (384). Let's finish putting everything in the fridge together! rigerator

**Alice:** Bob, I'm already at the fridge with <pudding> (383) and <apple> (385). Let's put the last items in the fridge together! rigerator

**Bob:** Alice, got <cupcake> (386) from the bedroom. Let's head to the fridge together to finish up! rigerator

**Alice:** Alice, let's bring the <pudding> (383) and <cupcake> (386) to the fridge (165) together! rigerator

**Bob:** Alice, heading to the fridge with <cupcake> (386). See you there! rigerator

**Alice:** Alice, let's grab the last <apple> (382) together and put both into the <fridge> (165)! rigerator

---

## B.4 STOP-TOKEN FAILURE AND SIMULATED DIALOGUE

Perhaps the most critical failure mode is **Multiple-Utterance Emission**. As shown in **Example B.5**, the model fails to recognize the stop token after its turn. Consequently, it begins to simulate the partner's response, hallucinating entire exchanges. This is characterized by the appearance of lowercase speaker tags (e.g., **alice:**) within a single generation block.

---

**Example B.4: Multiple-Utterance Emission within a Single Output**

**Bob:** Can you check the kitchen? I think the poundcake and wine might be there. I'll continue looking for the second cupcake in the living room. If you find anything, please let me know!

**Alice:** Found <poundcake> (372) and <fridge> (103) in the kitchen. I need your help to find 1 more cupcake and 1 apple. Let's work together to finish this task quickly! rigerator Let's split up - I'll look for the cupcake in the remaining cabinets and you check the living room for the apple. Once we have those, we can put them in the fridge together. **alice:** I'll check the remaining kitchencabinets for the cupcake, and you can search the living room for the apple. Let's meet back here to put everything in the fridge! **alice:** Found <cupcake> (368) in <kitchencabinet> (75). Heading to living room for the apple. **alice:** Found <apple> (101) in the living roo

**Bob:** Found <cupcake> (375) and <apple> (377) in the living room. Ready to put all items in the fridge with you! **alice:** Perfect! Let's head to the fridge together and put everything inside. Thanks for your help, Bob! **alice:** I'm already by the fridge. Let's combine our items: <poundcake> (372), <cupcake> (368), <cupcake> (375), <apple> (377), and <wine> (102). All set to be stored! **alice:** All items are now in the fridge. Great teamwork! **alice:** Great teamwork indeed! Let's take a quic

**Alice:** Let's store the remaining items: <cupcake> (375) and <wine> (102) in the fridge. I'm here and ready. **alice:** Placing <cupcake> (375) and <wine> (102) in the fridge. All done! **alice:** All items are now in the fridge. Great teamwork! **alice:** Moving on to the next task. See you around, Bob! **alice:** If you need help, feel free to reach out. Stay productive! **alice:** Let's keep the momentum going. Looking forward to our next task. **alice:** Great work! Let's continue with the next task. **alice:**

---

# C    ADDITIONAL COMPARISON OF GPT-SERIES MODELS ON C-WAH

To complement the main results in Table 1, we report a focused comparison of GPT-series backbones in the symbolic C-WAH setting. For each backbone, we evaluate both the CoELA architecture and our Mind-Map Agent, and measure (i) Average Steps (AS), (ii) the average number of API tokens consumed per episode (Tok/ep), and (iii) the average number of tokens per environment step (Tok/step). Token counts include both prompt and completion tokens over the entire episode and thus reflect the total computational budget required to run the agent.

Table 5: C-WAH symbolic observation results for GPT-series backbones. We report Average Steps (AS; lower is better), average tokens per episode (Tok/ep; lower is better), and average tokens per step (Tok/step; lower is better).

| Backbone | Agent | AS ↓ | Tok/ep ↓ | Tok/step ↓ |
|---|---|---|---|---|
| GPT-4 | CoELA | 60.0 | 429,693 | 7,162 |
| | Mind-Map Agent | **53.5** | 530,027 | 9,907 |
| GPT-4o-mini | CoELA | 65.0 | 561,596 | 8,640 |
| | Mind-Map Agent | **57.7** | 585,300 | 10,144 |
| GPT-5-mini | CoELA | 55.0 | 614,283 | 11,169 |
| | Mind-Map Agent | **49.0** | 1,740,848 | 35,528 |
| GPT-5-nano | CoELA | 70.0 | 1,364,227 | 19,489 |
| | Mind-Map Agent | **58.7** | 1,526,356 | 26,003 |

Across all four backbones, the Mind-Map Agent consistently reduces the number of environment steps required for successful completion relative to CoELA (Table 5, AS column). This holds for both inference models (GPT-4, GPT-4o-mini) and reasoning models (GPT-5-mini, GPT-5-nano), indicating that explicit structured reasoning can stabilize cooperative planning even when the underlying model capacity differs substantially. More broadly, the fact that the long-term memory mechanism improves performance across LLMs suggests that agents benefit from maintaining and incrementally refining a persistent representation of the task, rather than reconstructing plans independently at each decision point. This memory-centric view of reasoning appears to be a key design principle for effective cooperative embodied decision-making.

However, this gain in step efficiency comes with a non-negligible increase in token-level computational cost. Because the Mind-Map Agent performs two stages of reasoning at each decision point (Mind-Map Generation and Plan Selection), it naturally incurs more longer prompts and output tokens than the CoELA architecture. Even though we replace CoELA's explicit message-generation stage with the internal *MessageDraft* field inside the mind map, and do not rely on unconstrained chain-of-thought style reasoning, the input-output cost of generating and updating the mind map itself remains substantial. As a result, the average token consumption per episode is higher for the Mind-Map Agent across all GPT backbones, ranging from a modest overhead of about $1.0$–$1.2\times$ (GPT-4, GPT-4o-mini) to a substantial increase of nearly $2.8\times$ for GPT-5-mini. GPT-5-nano also shows a clear trade-off, reducing AS from $70.0$ to $58.7$ while increasing Tok/ep from $1.36M$ to $1.53M$, with corresponding increases in Tok/step.

These results clarify the computational trade-off underlying Mind-Map reasoning. In C-WAH, the Mind-Map Agent uses fewer interaction steps and fewer utterances (Section **??**), but this is achieved by shifting effort into structured internal reasoning and memory management, which is reflected in higher token counts. In safety-critical or high-stakes settings, such a shift from purely step-based efficiency toward more cautious, redundancy-reducing reasoning can be desirable; in resource-constrained deployments, the additional token budget must be explicitly accounted for. In our current implementation we deliberately allow relatively verbose mind-map outputs to expose the agent's plan selection process in an interpretable way. However, more efficient variants are possible: for example, by partially updating only the relevant parts of the mind map or triggering updates only when the task state or dialogue changes sufficiently, the Mind-Map reasoning could be realized with a smaller computational cost.

## D PROMPTS TEMPLATE

---

### Mind-Map Generation Prompt

Assume the role of my internal advisor, combining logical reasoning with cooperative decision-making strategies to optimize our shared objectives. Begin by analyzing $OPPO_NAME$'s inferred intentions based on our dialogue history and the progress we've made so far. Then, construct a structured mind map.

Below is an example of the expected mind map format:

```
{"InferredOthersIntentions":{
   "$OPPO_NAME$":[{"Task":"Place the wine and apple on the
coffeetable", "Reason": "$OPPO_NAME$ is holding the wine
and has committed to transporting apple together to the
coffeetable."}, {"Task": "Search for the pancake in the
bedroom", "Reason": "After placing the wine and apple,
$OPPO_NAME$ will explore the bedroom to locate the last
missing item, following the agreed plan."}]},
"CooperativeStrategy": {
   "CurrentAssignments":[{"Task":"Search the remaining
unchecked containers in the kitchen for the pancake.",
"AssignedTo":"$AGENT_NAME$"},{"Task": "Take the wine and
apple to the coffeetable, then explore the bedroom for the
pancake.", "AssignedTo": "$OPPO_NAME$"}],
   "LongTermPlan": {"Step 1": "[gocheck] stove (157) to find
the pancake.","Step 2-1":"[gograb] pancake if found in the
kitchen and place it on the coffeetable. It's last target
object.", "Step 2-2": "[gocheck] microwave (167), dishwasher
(159) to find the pancake.","Step 3": "[goput] pancake on the
coffeetable."},
   "ImmediateFocus": "Search the pancake and grab it if found
in the kitchen."},
"CommunicationStrategy": {"DialogueSummary": "...",
   "MessageDraft": "$OPPO_NAME$, I'll continue searching the
kitchen for the pancake while you transport the wine and apple
to the coffeetable. Let me know once you start exploring
the bedroom, so we can reassess if needed!", "ReasonsToSend":
"...", "ReasonsNotToSend": "..."}}
```

Now, using the given data, construct the mind map in JSON format. Keep the response strictly formatted without additional explanations.

Physical and Environmental Constraints:

- I can hold two objects at a time.
- All objects are denoted as <name>(id), such as <table>(712).
- Actions require multiple steps to complete.
- Moving between rooms or placing target objects at the goal location may be costly, so plan these actions sparingly.

Inputs: I'm $AGENT_NAME$. Given our shared goal, dialogue history, and my progress and previous actions, please help me generate a precise and concrete mind map in JSON format. This mind map should help me cooperate with $OPPO_NAME$ by guiding action planning and coordinating our efforts toward accomplishing our goal effectively.

Goal: $GOAL$

Progress: $PROGRESS$

Dialogue history:

$DIALOGUE_HISTORY$

Previous actions: $ACTION_HISTORY$

Previous Mind Map: $MIND_MAP$

Available actions: $AVAILABLE_ACTIONS$

$AGENT_NAME$'s Mind Map:

---