# OpenReview forum: "Mind-Map Agent: Enhancing Cooperative Task Planning through Communication Alignment with Large Language Models"
_ICLR.cc/2026/Conference — Submitted to ICLR 2026_

### Official Review · Reviewer_7MRf · 2025-10-30

**Soundness:** 2
**Presentation:** 3
**Contribution:** 2
**Rating:** 4
**Confidence:** 4

**Summary:**

The paper presents the Mind-Map Agent, an extension of the cooperative embodied agent CoELA, which introduces an explicit internal reasoning structure—the mind map—to help mitigate confabulation and reduce redundant dialogue in language-based cooperation effectively. Experiments conducted on C-WAH and TDW-MAT demonstrate that the proposed approach outperforms both CoELA and traditional baselines. An ablation study further confirms the contribution of each component within the Mind-Map framework.

**Strengths:**

- The paper identifies a critical limitation in communicative cooperative planning and proposes a well-motivated approach.
- The paper is well-written.
- The ablation experiments are commendable.

**Weaknesses:**

- The contribution appears incremental relative to existing works such as CoELA and CaPo.
- The paper lacks sufficient details on how the Mind-Map, the core component of the proposed method, is updated. Does this process involve additional LLM calls? If so, what are the efficiency trade-offs in terms of total computational cost and LLM usage?
- While evaluating the method across four different LLM backbones is commendable, the observation made in line 347 that Qwen2.5-7B cannot even follow the prompts to finish the task makes these results meaningless. Also, there is only one LLM-based baseline (CoELA) included. Incorporating additional baselines, such as CaPo mentioned in the paper, would strengthen the empirical validation. Moreover, as noted in line 357, testing CoELA with unadapted prompts and different LLM backbones from its original setup raises concerns about fairness and comparability. Evaluating the proposed method using the same backbone originally employed by the baseline would make the results more convincing.

**Questions:**

- What's the actual implementation of updating the Mind-Map? Is there LLM call required? How many?

- Why not test other existing baselines like CaPo?

- Why not report performance with the original llm backbone for CoELA?

---

> ### Author Response · Authors · 2025-11-26
>
> We sincerely thank you for the careful reading and constructive questions on implementation details, baselines, and fairness. We address each point below.
>
> (1) Mind-Map update mechanism and efficiency
>
> Each decision step in our method uses two LLM calls:
> 1. Mind-Map Generation: given the text interface (goal, dialogue, observations, action history) and the previous Mind-Map, the model outputs an updated Mind-Map in our JSON-style schema.
> 2. Plan Selection: conditioned on this updated Mind-Map plus the text interface, the model selects the next environment action or message.
>
> CoELA likewise uses two calls per step: one for message generation and one for plan selection. In Mind-Map Agents, we reuse the CoELA plan-selection prompt and _replace_ CoELA’s explicit message-generation stage with the MessageDraft field inside the Mind-Map, rather than adding any extra calls. Thus, the number of calls per step is unchanged; what differs is that the first call now maintains a structured control state instead of only proposing a surface-level utterance.
>
> Appendix C reports per-episode and per-step token usage for several GPT-series models. For GPT-4 and GPT-4o-mini, Mind-Map Agents reduce environment steps while increasing total tokens by roughly 1.2×, reflecting a deliberate trade-off: more LLM tokens are spent to obtain shorter trajectories and fewer redundant utterances, which can be acceptable when physical execution and sensing dominate cost. This trade-off is not uniform: for example, GPT-5-mini achieves strong step efficiency but with high token cost per step, as it tends to generate very verbose Mind-Maps. For such reasoning-oriented models, tighter prompt constraints or stronger control over Mind-Map verbosity will be important in future work.
>
> (2) Qwen2.5-7B behavior
>
> We agree that Qwen2.5-7B does not reliably follow prompts in our setting, and we do not present its absolute scores as competitive. We included it to show how a lightweight open model behaves under the same cooperative control scheme and to test whether Mind-Map structure still helps in this challenging regime.
>
> In symbolic C-WAH, CoELA+Qwen often behaves similarly to a Random Agent that samples among candidate plans, whereas Mind-Map+Qwen, despite noisy language and chattering, produces more meaningful cooperative dialogue and improves Average Steps. Since AS counts both physical actions and communication turns, this reflects a genuine difference in how planning and messaging are balanced.
>
> In the visual setting, all agents speak more because object observability is lower: initial viewpoints are randomized, so the same episode can vary greatly in difficulty depending on whether key objects are seen early or missed entirely, leading to large variance in steps. In this harder regime, the Mind-Map Agent still remains more robust than CoELA. We therefore view the Qwen2.5-7B results as informative stress tests for weaker backbones, highlighting where structured control helps and where current LLMs remain brittle.
>
> (3) Additional baselines
>
> For LLM-based baselines within the same modular architecture, we focused on CoELA because it shares perception, memory, communication, and planning modules, and because the CoELA framework provides ready-to-use C-WAH and TDW-MAT implementations. Classical MHP/RHP planners and a Random Agent act as non-LLM references.
> For CaPo, REVECA, and CoTS, we now include a dedicated discussion and an additional C-WAH comparison table in Appendix A. These methods typically assume centralized or semi-centralized coordination with shared internal information and free internal messages (e.g., synchronized meta-planning, validation over collaborator trajectories, shared search trees). In our setting, each agent controls its own embodiment and every natural-language message is explicitly treated as an environment step. A direct port of CaPo/REVECA/CoTS would therefore either violate this assumption or give them strictly stronger capabilities. Appendix A contrasts centralized (REVECA, CoTS) and decentralized (CoELA, Mind-Map) regimes and separates environment steps from “free” internal messages; we restrict our empirical claims to the decentralized, costed-communication setting.
>
> (4) CoELA backbones, GPT-4, and fairness
>
> We share the reviewer’s concern about fairness under changing backbones. In the revision, we therefore include GPT-4 results (the original CoELA backbone) in Appendix C. However, GPT-4 is costly and subject to API lifecycle changes, which may hinder long-term reproducibility. This is why we also report results with more recent GPT variants and with open models such as Llama3.3-70B and Qwen2.5-7B: our aim is to make it easier for future work to reproduce and extend our findings even if specific proprietary backbones become less accessible.
>
> We hope these clarifications on the update mechanism, efficiency trade-offs, and baseline choices address the reviewer’s concerns.

---

### Official Review · Reviewer_A9Ey · 2025-10-31

**Soundness:** 2
**Presentation:** 3
**Contribution:** 2
**Rating:** 4
**Confidence:** 4

**Summary:**

The paper introduces **Mind-Map Agent**, a framework designed to enhance cooperative task planning and communication in multi-agent settings. The key idea is to guide LLM-based reasoning with a **structured memory system**, called the *Mind-Map*, which explicitly separates others’ intentions, cooperative strategy, and communication strategy. This structure aims to reduce confabulation, maintain consistency in long-horizon reasoning, and lower communication overhead during collaboration.

The authors evaluate the proposed framework on two embodied AI benchmarks, C-WAH and TDW-MAT**,** using multiple LLMs of different scales. Results show that the Mind-Map Agent generally achieves higher cooperative efficiency and fewer communication turns compared to baseline methods such as COELA and classical planners. An ablation study further analyzes the contribution of each Mind-Map component.

**Strengths:**

- The paper is well-written and easy to follow, raising important unsolved issues in multi-agent communication and long-horizon plan consistency.
- By introducing an inspectable memory schema, the work moves toward making language-agent reasoning more explainable. The Mind-Map entries (intentions, cooperative strategies, communication plans) allow researchers to trace decision-making steps and understand more about how LLM agents communicate with each other.

**Weaknesses:**

- The framework of the Mind-Map Agent largely builds upon the existing COELA architecture, with the primary modification being the introduction of a structured memory system. While this addition is interesting, the overall contribution appears somewhat limited in scope. The concept of structured or persistent memory has already been explored in prior embodied agent systems, such as WAH [4], where mechanisms like MCTS were used to record information and guide decision-making.
- The experimental comparison is currently insufficient to substantiate the claimed improvements. The evaluation primarily contrasts Mind-Map with COELA, omitting more recent and relevant baselines such as REVECA [1], CaPo [2], and CoTS [3], which have advanced the design of collaborative embodied agents after COELA. Although some of these works are cited in the related work section, quantitative or qualitative comparisons are missing.
- It remains unclear whether the Mind-Map Agent can effectively cooperate with heterogeneous partners, including agents of different architectures or human collaborators. Since the proposed system relies on internally inferred intentions and self-developed cooperative strategies, it is uncertain how well these inferences align with partners that follow different reasoning paradigms.

*[1]* Seo, SeungWon, et al. "Reveca: Adaptive planning and trajectory-based validation in cooperative language agents using information relevance and relative proximity." *Proceedings of the AAAI Conference on Artificial Intelligence*. Vol. 39. No. 22. 2025.

*[2] Liu, Jie, et al. "CaPo: Cooperative Plan Optimization for Efficient Embodied Multi-Agent Cooperation." The Thirteenth International Conference on Learning Representations.*

*[3] Zu, Lizheng, et al. "Collaborative Tree Search for Enhancing Embodied Multi-Agent Collaboration." Proceedings of the Computer Vision and Pattern Recognition Conference. 2025.*

*[4] Puig, Xavier, et al. "Watch-And-Help: A Challenge for Social Perception and Human-AI Collaboration." International Conference on Learning Representations.*

**Questions:**

- See Weaknesses.
- Providing examples that illustrate the evolution trajectories of the Mind-Map memory throughout task progression would greatly help readers understand how effectively the memory system supports long-horizon planning. In particular, case studies showing how the memory representation changes in response to a partner’s messages or environmental updates could clarify the dynamic behavior and interpretability of the proposed structured memory.
- The ablation results in Table 3 indicate that the Communication Strategy alone achieves notably strong performance compared to more complex combinations of components. Regarding this, the author mentions that flexibility is important. However, the design of Mind-Map is a structured schema. This raises the question of whether certain structural constraints or components of the Mind-Map might be unnecessary—or even detrimental—to overall performance.

---

> ### Author Response · Authors · 2025-11-26
>
> We sincerely thank you for your detailed feedback. Your insightful comments have prompted additional analyses and helped sharpen our study. We address each of your points in detail below.
>
> (1) Positioning relative to CoELA and prior memory work
>
> Mind-Map Agents build on the CoELA architecture, and persistent or structured memory has already been explored in embodied systems such as WAH with MCTS-based tracking. Our goal is not to introduce long-term memory itself, but to study what happens when Theory-of-Mind–style components (others’ intentions, cooperative strategy, communication strategy) are turned into an explicit control state that the LLM must maintain and reuse during a cooperative task.
> In many memory-augmented LLM agents, the model retrieves past dialogue or state summaries, but multi-step plans remain implicit in chain-of-thought and are discarded at the next step; new observations can break consistency or even cause the agent to be steered by its own previous messages. In Mind-Map Agents, by contrast, the LLM explicitly maintains and updates a compact internal state about partner behavior, work division, communication necessity, and priority between partner requests and its own tasks, and then uses this state for plan selection. The contribution is therefore in how memory is structured and used for control, rather than in the existence of long-term memory itself.
>
> (2) Experimental comparison and baselines
> Our quantitative comparison mainly contrasts Mind-Map vs. CoELA and classical planners, but we do so across several LLM families and scales: in addition to GPT-series models, we include Llama3.3-70B, GPT-OSS-120B, and Qwen2.5-7B under the same CoELA vs. Mind-Map setup, and observe that the Mind-Map schema stabilizes cooperative behavior.
> For recent cooperative agents such as REVECA, CaPo, and CoTS, assume centralized or semi-centralized coordination with shared internal information and free internal messages, whereas in our setting each agent controls its own embodiment and every natural-language message consumes an environment step. Appendix A therefore contrasts centralized (REVECA, CoTS) and decentralized (CoELA, Mind-Map) settings and separates environment steps from internal messages.
>
> (3) Heterogeneous partners and human collaboration
> We acknowledge that the current experiments do not fully cover heterogeneous partners or human collaborators. In this work, human partners were replaced by LLM agents to enable controlled comparison, which is not a substitute for user studies. Given the small number of official test episodes and the previous web-based, non-visual interface (where the other agent’s behavior is only visible via messages), we judged that a small-scale human study would not yet yield robust conclusions.
> Mind-Map Agents can already cooperate with heterogeneous non-human partners, such as an MCTS-based agent that uses template messages and a CoELA-style agent that uses natural language. We are running additional experiments in these heterogeneous collaboration setups; if completed in time, we will include the comparison plots, and otherwise we will list the absence of heterogeneous- and human-partner evaluation as an explicit limitation.
>
> (4) Mind-Map evolution trajectories
> We agree that concrete Mind-Map trajectories would clarify how the memory supports long-horizon planning. In the revision, we will add a case study in the appendix showing how the Mind-Map evolves over key steps in a C-WAH episode, including updates after partner messages and unexpected observations. We are also preparing visualizations and short videos for the project page so that readers can inspect Mind-Map updates alongside the environment rollout.
>
> (5) Ablation results and the role of structure
> Regarding Table 3, the current ablation relaxes schema constraints, and the in-context examples for reduced variants are sparse. When only a single component is prompted—especially the Communication Strategy—GPT-4o-mini often “fills in” missing parts of the Mind-Map (e.g., intentions or long-term plans) by itself. This can improve performance but blurs the boundary between components and makes it difficult to attribute gains to a specific field. To obtain a clearer picture, we are running additional ablations in which the same schema constraints remain active even when subsets of components are used, so that the incremental effect of each part can be measured under consistent structural assumptions. We will update Table 3 and the corresponding discussion accordingly.

---

### Official Review · Reviewer_TfWU · 2025-10-31

**Soundness:** 3
**Presentation:** 2
**Contribution:** 2
**Rating:** 4
**Confidence:** 2

**Summary:**

This paper proposes Mind-Map Agent, an LLM-driven framework that separates (i) dialogue history and working memory from (ii) a structured long-term “Mind-Map” (intentions, cooperative strategy, and communication strategy). The system reasons in two phases: Mind-Map Generation and Plan Selection, and aims to mitigate confabulation, reduce redundant dialogue, and improve long-horizon coordination. Experiments on C-WAH and TDW-MAT (with symbolic and visual observations, several LLM backbones) show consistent reductions in steps and often communication turns versus CoELA and classical planners. An ablation on the schema’s components is also presented.

**Strengths:**

* Clear, motivated idea: Explicit, inspectable schema for intentions and strategies is well aligned with Theory of Mind literature and practical multi-agent needs.
* General mechanism: The JSON-style schema and two-phase prompting is architecture-agnostic and easy to port across models and environments.
* Compelling empirical signal: Across multiple backbones and two benchmarks, Mind-Map variants typically achieve notably fewer steps than CoELA and classical baselines; communication turns often drop substantially as well.
* Interpretability: The design encourages transparent, auditable reasoning (e.g., fields like “Reason to Send” vs “Reason not to Send”), which is valuable for human-robot interaction.
* Thoughtful discussion: The paper acknowledges trade-offs such as communication efficiency versus cautious reasoning.

**Weaknesses:**

1. Evaluation scope and baselines

   * Only two simulation benchmarks are used, and both are from the CoELA framework. Protocolized baselines (e.g., CaPo, REVECA) are discussed but not included. This weakens generality claims.
   * CoELA prompts were optimized for GPT-4, possibly disadvantaging other LLMs. Prompt engineering effort should be balanced across methods.

2. Metrics and statistical rigor

   * C-WAH evaluation only reports Average Steps (AS) and Average Communication Steps (ACS), with no explicit success rate. How failures are treated is unclear.
   * Only three runs are used per experiment, and no significance tests are provided. Variances are large in places.

3. Confabulation reduction is argued but not measured

   * The paper highlights that Mind-Map mitigates confabulation, but provides no quantitative measure or error count to support this.

4. Ablation clarity

   * Section 4.3 omits the structural key-matching process, which makes it hard to assess the true incremental contribution of each component.

5. Method specifics and reproducibility

   * More detail is needed: exact schema template, in-context examples, prompt templates for both reasoning phases, memory retrieval and pruning rules, token budgets, and compute details.

**Questions:**

Check weaknesses please.

---

> ### Author Response · Authors · 2025-11-26
>
> We sincerely thank the reviewer for the constructive feedback and for their positive assessment of the motivation, interpretability, and empirical results of Mind-Map Agents. We address the main concerns below.
>
> (1) Evaluation scope and baselines
>
> We agree that our empirical scope is limited to two simulation benchmarks, both built on the CoELA framework. We chose C-WAH and TDW-MAT because they differ in observation modality (symbolic vs. visual), coordination structure (watch-and-help vs. container-based transport), and difficulty, and because they are widely used embodied cooperation benchmarks with open implementations. In parallel, we are running larger-scale experiments in additional environments with more diverse task structures and more complex human–robot interaction; if these are completed in time for the camera-ready version, we plan to include them as additional validation.
>
> Protocolized baselines such as CaPo, REVECA, and CoTS rely on centralized or semi-centralized coordination with shared internal information and free internal messaging, whereas in our setting each agent controls its own embodiment and every natural-language message is counted as an environment step. A direct import would either violate these assumptions or give such methods strictly stronger capabilities. To clarify this, Appendix A includes an additional C-WAH comparison table that contrasts centralized approaches (REVECA and CoTS, using their reported settings with free direct messages) with decentralized CoELA and Mind-Map Agents, and explicitly separates environment steps from internal messages.
>
> To address the concern about possible GPT-4 bias, Appendix C reports a comparison across several GPT-series LLMs, including both step metrics and token usage. Mind-Map Agents consistently reduce steps across GPT backbones while incurring an increase in tokens due to the additional reasoning phase.
>
> (2) Metrics, variance, and statistical rigor
>
> In C-WAH, all evaluated agents (including the Random agent that samples among candidate plans) achieve 100% task success within the 250-step limit on the test episodes. The benchmark is therefore used as an efficiency testbed where the main signal is how many steps and communication turns are needed to reach the goal.
>
> We acknowledge that the number of runs per configuration is small. Each run evaluates the 10 test episodes once, and we repeat this three times. To the best of our knowledge, prior work typically reports a single run on these 10 episodes due to LLM cost; we are the first to report repeated runs and standard deviations. Because the 10 episodes differ substantially in the number of target objects and rooms, the per-episode difficulty is heterogeneous, which naturally produces relatively large variance in average steps.
>
> (3) Confabulation reduction
>
> We agree that our current evidence for confabulation reduction is indirect. In our experiments, confabulation typically appears as redundant or inconsistent dialogue that does not cause immediate failure but accumulates and lengthens trajectories. This makes it difficult to define a clean error rate on individual utterances or actions, and building a high-quality annotation protocol for long multi-agent dialogues would require substantial effort. In this work, we therefore rely on two partial signals: (i) reductions in steps and communication turns, and (ii) qualitative analyses of typical failure modes under CoELA (Appendix B). We see more direct measurement of confabulation as an important open problem. For example, LLM-as-a-judge style evaluations of dialogue plausibility, such as Collaborative Reasoner [1], could be adapted to our setting to quantify confabulation, but developing and validating such an evaluation pipeline is beyond the scope of this paper.
>
> (4) Ablation clarity
>
> We appreciate the concern about the ablation in Section 4.3. In the ablation study, the schema constraints are relaxed so that the model can freely generate subsets of the Mind-Map components. We agree that it is more important to understand how using the full, more constrained schema affects plan selection and stability. We are running additional experiments that keep these schema constraints active in the ablation setting and will update the table and discussion to reflect their incremental effect.
>
> (5) Method specifics and reproducibility
> Our agents invoke the Mind-Map Generation prompt with a single schema example. In the example, the model is instructed to generate a draft message inside the Mind-Map, which replaces CoELA’s separate message-generation stage. Plan selection uses the CoELA prompt with only an additional field that provides the current Mind-Map. For fair comparison and reproducibility, we include the Mind-Map Generation prompts in the appendix and plan to release official code.
>
> [1] Collaborative Reasoner: Self-Improving Social Agents with Synthetic Conversations, Ni et al., NeurIPS 2025

---

### Official Review · Reviewer_e7m7 · 2025-11-01

**Soundness:** 3
**Presentation:** 2
**Contribution:** 2
**Rating:** 4
**Confidence:** 4

**Summary:**

This paper introduce Mind-Map Agent, a LLM-based structured reasoning framework for multi-agent embodied AI tasks. The key idea is to prevent confabulation and excessive dialogue in LLM-based cooperative planning by introducing a Mind-Map memory, which is a structured long-term representation that explicitly separates dialogue history, task state, inferred intentions, cooperative strategy, and communication strategy. Experiments on C-WAH and TDW-MAT demonstrate that the proposed framework achieves better performance and costs fewer dialogue turns, better task completion than CoELA and MCTS-based baselines.

**Strengths:**

1. Mind-Map agent is an interpretable and effective way to structure LLM reasoning in cooperative tasks.
2. Experiments includes multiple models (GPT-4o-mini, GPT-OSS-120B, Llama 3.3-70B, Qwen2.5 7B) and two benchmarks, showing consistent performance gains and reduced communication rounds.
3. Ablation studies demonstrates how each Mind-Map component contributes differently to efficiency.

**Weaknesses:**

1. Although emergent behaviors are mentioned, richer qualitative visualization or human evaluation are needed.
2. While Mind-Map reasoning reduces hallucination and dialogue turns, it can increase computational cost with more LLM queries and memory management.
3. The proposed Mind-Map is essentially a reorganization of existing memory modules. Its “Long-Term Memory” resembles mechanisms already explored in LLM-based agents such as Long-Term Memory: The Foundation of AI Self-Evolution, while the maintenance of cooperative strategies are proposed by CaPo (ICLR 2025), and the representation of intentions and dialogue history are proposed from CoELA (ICLR 2024). This paper mainly combines these memory elements into a unified JSON format.

**Questions:**

See weaknesses

1. Could you provide more qualitative examples, including the evolution of Mind-Map over steps?
2. Could you add more baselines, such as CaPo, REVECA, and CoTS?

CaPo: Cooperative Plan Optimization for Efficient Embodied Multi-Agent Cooperation. ICLR 2025

Collaborative Tree Search for Enhancing Embodied Multi-Agent Collaboration. CVPR 2025

REVECA: Adaptive Planning and Trajectory-based Validation in Cooperative Language Agents using Information Relevance and Relative Proximity. AAAI 2025

---

> ### Author Response · Authors · 2025-11-26
>
> We sincerely thank you for the constructive feedback and recognizing the interpretability of Mind-Map Agents, the breadth of our evaluation, and the usefulness of the ablations. We address the main points below and will add the corresponding clarifications in the revision.
>
> (1) Positioning of Mind-Map Agents and Relation to Prior Memory Work
>
> We agree that long-term memory for agents is well studied, and we do not claim novelty in using a persistent memory itself. Our focus is on how this memory is structured and used for control. The Mind-Map schema organizes three Theory-of-Mind–inspired components (inferred others’ intentions, cooperative strategy, communication strategy) into an explicit, LLM-readable control state that is maintained and updated throughout a cooperative embodied task. Rather than logging past events, this state is what the LLM consults when deciding both actions and messages under a limited step budget. This perspective complements CoELA, which mainly relies on step-by-step reactive reasoning without a persistent cooperative control state, and protocolized meta-planning approaches such as CaPo, which rely on more rigid and centralized discussion procedures. Recent work on long-term memory has emphasized that a key open problem is making LTM genuinely useful in complex, interactive multi-agent settings [1]. Our contribution is to provide one concrete embodied instantiation of this direction: agents must manage a structured long-term memory while collaborating through language under partial observability and communication cost. Our experiments indicate that Mind-Map Generation, i.e., repeatedly updating and using this structured control state, reduces LLM-induced chattering and confabulation and improves cooperative efficiency across several LLM families and model scales.
>
> (2) Qualitative Examples and Computational Cost
>
> We agree that richer qualitative analysis was needed. In the revision we will add a multi-step Mind-Map trajectory for a representative episode, illustrating how inferred intentions, cooperative strategy, and communication strategy evolve over time and affect action or message choices. In addition, Appendix B now describes LLM-induced “chattering” under CoELA with concrete dialogue examples (repetition loops, agreement loops, convergence loops, multi-utterance hallucinations), which the Mind-Map design helps suppress. On computation, we now report token usage per episode and per step for several GPT backbones (Appendix D). Mind-Map Agents consistently reduce Average Steps compared to CoELA, while incurring a increase in total tokens due to the Mind-Map Generation phase. In embodied settings where physical execution and sensing usually dominate latency and cost, we view this as an explicit trade-off: a larger reasoning budget in exchange for shorter trajectories, fewer redundant utterances, and more stable cooperative behavior.
>
> (3) Baselines: CaPo, REVECA, and CoTS
>
> We appreciate the suggestion to include CaPo, REVECA, and CoTS and agree that clearer positioning is important. After examining their implementations, we found that their core assumptions differ from our decentralized, costed-communication setting, making a direct comparison non-trivial. CaPo assumes a meta-planner that can coordinate agents through centralized discussion rounds. REVECA and CoTS introduce centralized or shared reasoning structures (e.g., validation over collaborator trajectories, shared search trees) and rely on direct messaging channels with guaranteed delivery that do not incur step cost. In our C-WAH and TDW-MAT setup, each agent controls its own embodiment, cannot access teammates’ internal logs, and every natural language message is counted as an environment step. Porting CaPo/REVECA/CoTS without modification would therefore give them strictly stronger information and communication than our agents, while adapting them to our constraints would require substantial algorithmic changes and would no longer represent the original methods. To clarify this, the revision adds a short theoretical framing and comparison (Appendix A) that categorizes these methods as centralized controllers with shared information states and free internal communication. We also report a simple quantitative comparison that distinguishes between environment steps and internal messages, and show that Mind-Map Agents achieve competitive or better effective efficiency without access to a free channel. We hope these clarifications address the reviewer’s concerns.
>
> [1] Long-Term Memory: The Foundation of AI Self-Evolution, arXiv:2410.15665

---

### Meta-Review · Area_Chair_mnU4 · 2026-01-06

**Summary:**

Summary:
This paper introduces Mind-Map Agent, a framework for LLM-based multi-agent tasks. It uses a Mind-Map memory to separate dialogue history from task state, agent intentions, and communication plans. This helps reduce hallucination and unnecessary dialogue, and keeps planning consistent. Experiments on C-WAH and TDW-MAT show that Mind-Map Agent completes tasks with fewer steps and fewer dialogue turns than CoELA.

Strengths:
1. The method is interpretable and shows good performance on two benchmarks.
2. The ablation studies demonstrate how each component contributes to efficiency.
3. The idea is clear and well-motivated.

Weaknesses (initial reviewer concerns):
1. Lack of qualitative visualization or human evaluation.
2. The computational costs could be high.
3. The proposed method combines multiple existing methods.
4. CoELA was optimized for GPT-4, hence the comparison might not be fair.
5. Evaluation lacks rigor (no success rate for C-WAH and only 3 runs).
6. No quantitative measure for the migration of confabulation.
7. Hard to assess the true incremental contribution of each component.
8. Missing method details.
9. Missing important baselines such as REVECA, CaPo, and CoTS.
10. Unclear whether the method can effectively cooperate with heterogeneous partners, including humans.

**Reviewer Concerns:**

Concerns 4, 5, 6, 7, and 8 were adequately addressed. The responses to other concerns were not sufficiently convincing. In particular, the lack of comparison against more recent baselines, limited novelty (compared to CoELA and other more recent work on memory), and the lack of evaluation with humans are important concerns that have not been adequately addressed.

**Reviewer Scores:**

Reviewer e7m7 is not very likely to raise the score since the concerns about the cost and novelty still remain valid.

Reviewer TfWU might raise their score, since the responses seem to address their concerns.

Reviewer A9EY’s concerns about novelty and lack of evaluation with real humans have not been convincingly addressed, so it is unlikely that they would raise the score.

Similarly, Reviewer 7MRf’s concern about novelty remains insufficiently addressed, so they are unlikely to raise the score.

---

### Decision · Program_Chairs · 2026-01-26

Reject